# Self-Critical Reasoning
# for Robust Visual Question Answering

**Jialin Wu**
Department of Computer Science
University of Texas at Austin
jialinwu@utexas.edu

**Raymond J. Mooney**
Department of Computer Science
University of Texas at Austin
mooney@cs.utexas.edu

## Abstract

Visual Question Answering (VQA) deep-learning systems tend to capture superficial statistical correlations in the training data because of strong language priors and fail to generalize to test data with a significantly different question-answer (QA) distribution [1]. To address this issue, we introduce a self-critical training objective that ensures that visual explanations of correct answers match the most influential image regions more than other competitive answer candidates. The influential regions are either determined from human visual/textual explanations or automatically from just significant words in the question and answer. We evaluate our approach on the VQA generalization task using the VQA-CP dataset, achieving a new state-of-the-art *i.e.*, 49.5% using textual explanations and 48.5% using automatically annotated regions.

## 1 Introduction

Recently, Visual Question Answering (VQA) [4] has emerged as a challenging task that requires artificial intelligence (AI) systems to compute answers by jointly analyzing both natural language questions and visual content. The state-of-the-art VQA systems [8, 2, 1, 3, 12, 33, 25, 29, 14, 15, 21] achieve high performance when the training and test question-answer (QA) pairs are sampled from the same distribution. However, most of these systems fail to generalize to test data with a substantially different QA distribution. In particular, their performance drops catastrophically on the recently introduced Visual Question Answering under Changing Priors (VQA-CP) [1] dataset. The strong language priors encourage systems to blindly capture superficial statistical correlations in the training QA pairs and simply output the most common answers, instead of reasoning about the relevant image regions on which a human would focus. For example, since about 40% of questions that begin with "what sport" have the answer "tennis", systems tend to learn to output "tennis" for these questions regardless of image content.

A number of recent VQA systems [28, 35, 25, 20] learn to not only predict correct answers but also be "right for the right reasons" [23, 25]. These systems are trained to encourage the network to focus on regions in the image that humans have somehow annotated as important (which we will refer to as "important regions."). However, many times, the network also focuses on these important regions even when it produces a wrong answer. Previous approaches do nothing to actively discourage this phenomenon, which we have found occurs quite frequently.[1] For example, as shown in Figure 1, we ask the VQA system, "What is the man eating?". The baseline system predicts "hot dog" but focuses on the banana because hot dog appears much more frequently in the training data. What's worse, this error is hard to detect when only analyzing the correct answer "banana" that has been successfully grounded in the image.

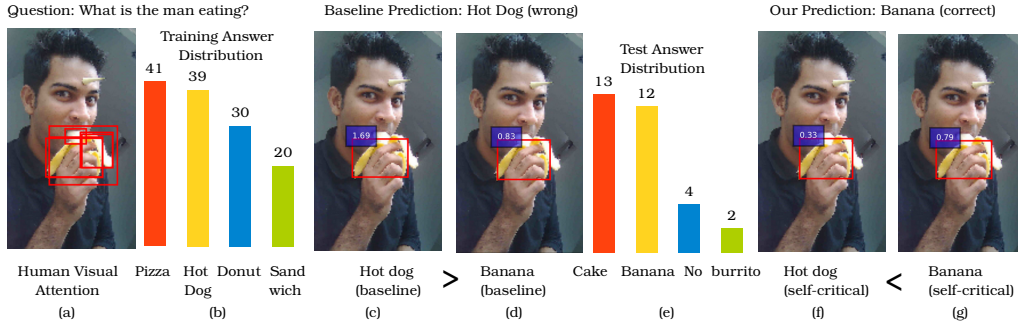

Figure 1: Example of a common answer misleading the prediction even though the VQA system has the right reasons for the correct answer. Figure (a) shows the important regions extracted from human visual attention. Figure (b), (e) show the answers' distribution for the question "What is the man eating?" in the training and test dataset. Figure (c), (d) show the most influential region for the prediction "hot dog" and "banana" using the baseline UpDn VQA system and Figure (f), (g) show the influential region for the prediction "hot dog" and "banana" using the VQA system after being trained with our self-critical objective. The number on the bounding box shows the answer's sensitivity to the object.

To address this issue, we present a "self-critical" approach that directly criticizes incorrect answers' sensitivity to the important regions. First, for each QA, we determine the important region that most influences the network's prediction of the correct answer. We then penalize the network for focusing on this region when its predicted answer for this question is *wrong*.

Our self-critical approach is end-to-end trainable and only requires that the base VQA system be differentiable to the visual content, and thus can be applied to most current state-of-the-art systems. We investigated three approaches to determining important regions. First, like the previous work [28, 35, 25, 20], we used regions that humans have explicitly marked as important. However, this requires a fair bit of extra human effort to provide such detailed annotations. So we also explored using human textual VQA explanations from the VQA-X [18] dataset to determine important objects which are then grounded to important regions in the image. Finally, we tried determining important regions by only using objects mentioned in the question or answer and grounding them in the image, which requires *no* additional human annotation of the VQA training data.

We evaluate our approach using the UpDn VQA system [2] on the VQA-CP dataset [1] and achieve a new state-of-the-art performance (currently 47.7%): *i.e.* 49.5% overall score with VQA-X [18] textual explanations, 49.1 % with VQA-HAT [7] visual explanations and 48.5% using just mentioned objects in the questions and answers. Our code is available at `https://github.com/jialinwu17/Self_Critical_VQA`.

## 2   Related Work

### 2.1   Human Explanations for VQA

There are two main kinds of human explanations available for the most popular VQA dataset [4], *i.e.*, visual and textual explanations. The VQA-HAT dataset [7] is a visual explanation dataset that collects human attention maps by giving human experts blurred images and asking them to determine where to deblur in order to answer a given visual question. Alternatively, [18] presents the VQA-X dataset that associates a textual explanation with each QA pair, which a human has provided to justify an answer to a given question. In this work, we utilize both of these kinds of explanations to provide the important regions.

### 2.2   Language Priors in VQA

Language priors [1, 9] in VQA refer to the fact that question types and their answers are highly correlated. For instance, questions that begin with "How many" are usually answered by either two

or three. These language priors allow VQA systems to take a shortcut when answering questions by only focusing on the questions without reasoning about the visual content. In order to prevent this shortcut, VQA v2 [4] balances the answer distribution so that there exist at least two similar images with different answers for each question. Recently, [1] introduce a diagnostic reconfiguration of the VQA v2 dataset called VQA-CP where the distribution of the QA pairs in the training set is significantly different from those in the test set. Most state-of-the-art VQA systems are found to highly rely on language priors and experience a catastrophic performance drop on VQA-CP. We evaluate our approach on VQA-CP in order to demonstrate that it generalizes better and is less sensitive to distribution changes.

## 2.3   Improving VQA using Human Explanations

The desired property for VQA systems is to not only infer the correct answers to visual questions but also base the answer on image regions that a human believes are important, $i.e.$, right for the right reasons. The VQA systems that address this issue can be classified into two categories. The first trend is to build a system whose model is inherently interpretable. For example, GVQA [1] explicitly disentangles the vision and language components by introducing a separate visual concept verifier and answer cluster classifiers. The other trend is to align a systems' explanation to human experts' explanations for the correct answers. [35, 20] align the internal attention weights over the image to the human attention maps. The work most related to ours is HINT [25], which enforces the system's gradient-based importance scores for each detected object to have the same rankings as its human importance scores. In contrast to prior work, our approach not only encourages the systems to be sensitive to the important regions identified by humans, but also decrease the incorrect answers' sensitivity to these regions.

# 3   Preliminaries

In this section, we first introduce our base Bottom-up Top-down (UpDn) VQA system[2][2]. Then, we describe our method for constructing a proposed object set that covers the most influential objects on which a human would focus when answering the question.

## 3.1   Bottom-Up Top-Down VQA

A large number of previous VQA systems [8, 5, 21] utilize a trainable Top-Down attention mechanism over convolutional features to recognize relevant image regions. [2] introduced complementary bottom-up attention that first detects common objects and attributes so that the top-down attention can directly model the contribution of higher-level concepts. This UpDn approach is heavily used in recent work [25, 31, 14, 32, 26] and significantly improves VQA performance.

Technically, on the vision side, for each image, UpDn systems first extract a visual feature set $\mathcal{V} = \{\mathbf{v}_i, ..., \mathbf{v}_{|\mathcal{V}|}\}$ for each image whose element $\mathbf{v}_i$ is a feature vector for the $i$-th detected object. On the language side, UpDn systems sequentially encode each question $Q$ to produce a question vector $\mathbf{q}$ using a standard single-layer GRU [6] denoted by $h$, $i.e.$ $\mathbf{q} = h(Q)$. Let $f$ denote the answer prediction operator that takes both visual features and question features as input and predicts the confidence for each answer $a$ in the answer candidate set $\mathcal{A}$, $i.e.$ $P(a|\mathcal{V}, Q) = f(\mathcal{V}, \mathbf{q})$. The VQA task is framed as a multi-label regression problem with the gold-standard soft scores as targets in order to be consistent with the evaluation metric. In particular, the standard binary cross entropy loss $\mathcal{L}_{vqa}$ is used to supervise the sigmoid-normalized outputs.

## 3.2   Proposed Influential Object Set Construction

Our approach ideally requires identifying important regions that a human considers most critical in answering the question. However, directly obtaining such a clear set of influential objects from either visual or textual explanations is hard, as the visual explanations also highlight the neighbor objects around the most influential one, and grounding textual explanations in images is still an active research field. We relax this requirement by identifying a proposed set of influential objects $\mathcal{I}$ for each QA pair. This set may we noisy and contain some irrelevant objects, but we assume that it at

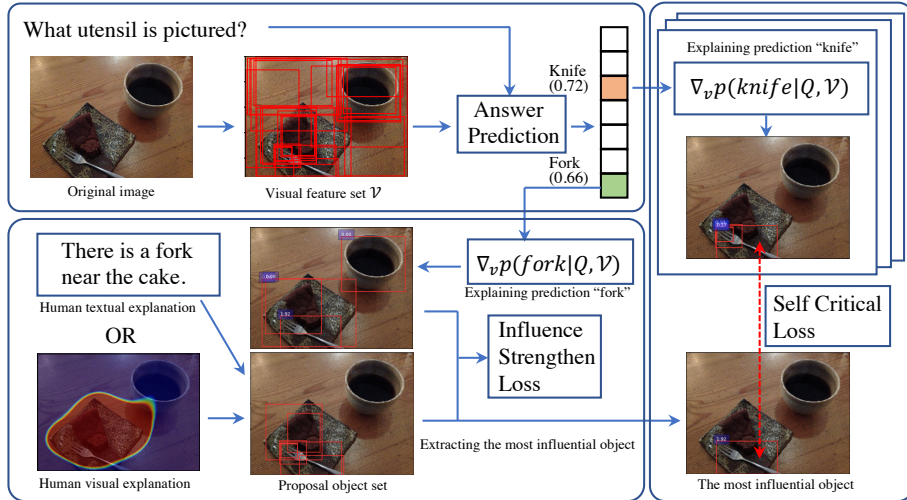

Figure 2: Model overview. In the left top block, the base UpDn VQA system first detects a set of objects and predicts an answer. We then analyze the correct answer's sensitivity (Fork) to the detected objects via visual explanation and extract the most influential one in the proposal object set as the most influential object, which is also further strengthened via the influence strengthen loss (left bottom block). Finally, we analyze the competitive incorrect answers' sensitivities (Knife) to the most influential object and criticize the sensitivity until the VQA system answers the question correctly (right block). The number on a bounding box is the answer's sensitivity to the given object.

least includes the most relevant object. As previously mentioned, we explore three separate methods for constructing this proposal set, as described below:

**Construction from Visual Explanations**. Following HINT [25], we use the VQA-HAT dataset [7] as the visual explanation source. HAT maps contain a total of $59,457$ image-question pairs, corresponding to approximately 9% of the VQA-CP training and test set. We also inherit HINT's object scoring system that is based on the normalized human attention map energy inside the proposal box relative to the normalized energy outside the box. We score each detected object from the bottom-up attention and build the potential object set by selecting the top $|\mathcal{I}|$ objects.

**Construction from Textual Explanations**. Recently, [18] introduced a textual explanation dataset that annotates $32,886$ image-question pairs, corresponding to 5% of the entire VQA-CP dataset. To extract the potential object set, we first assign part-of-speech (POS) tags to each word in the explanation using the spaCy POS tagger [11] and extract the nouns in the sentence. Then, we select the detected objects whose cosine similarity between the Glove embeddings [19] of their category names and any of the extracted nouns' is greater than $0.6$. Finally, we select the $|\mathcal{I}|$ objects with the highest similarity.

**Construction from Questions and Answers**. Since the above explanations may not be available in other datasets, we also consider a simple way to extract the proposal object set from just the training QA pairs alone. The method is quite similar to the way we construct the potential set from textual explanations. The only difference is that instead of parsing the explanations, we parse the QA pairs and extract nouns from them.

## 4   Approach

In this section, we present our self-critical approach to prevent the most common answer from dominating the correct answer given the proposal sets of influential objects. Figure 2 shows an overview of our approach. Besides the UpDn VQA system (left top block), our approach contains two other components, we first recognize and strengthen the most influential objects (left bottom block), and then we criticize incorrect answers that are more highly ranked than the correct answer and try to make them less sensitive to these key objects (right block). As recent research suggests that

gradient-based methods more faithfully represent a model's decision making process [25, 34, 30, 13], we use a modified GradCAM [24] to compute the answer $a$'s sensitivity to the $i$-th object features $\mathbf{v}_i$ as shown in Eq. 1.[3]

$$\mathcal{S}(a, \mathbf{v}_i) := \left(\nabla_{\mathbf{v}_i} P(a|V, q)\right)^T \mathbf{1} \tag{1}$$

There are two modifications to GradCAM: (1) ReLU units are removed, (2) gradients are no longer weighted by their feature vectors. This is because negative gradients on the inputs to a ReLU are valuable evidence against the current prediction. Therefore, there is no need to zero them out with a ReLU. Also, before they are weighted by the feature vectors, the gradients indicate how small changes in any direction influence the final prediction. If weighted by the feature vectors, the output tends to reflect the influence caused *only* by *existing* attributes of the objects, thereby ignoring other potential attributes that may appear in the test data.

## 4.1 Recognizing and Strengthening Influential Objects

Given a proposal object set $\mathcal{I}$ and the entire detected object set $\mathcal{V}$, we identify the object that the correct answer is most sensitive to and further strengthen its sensitivity. We first introduce a sensitivity violation term $\mathcal{SV}(a, \mathbf{v}_i, \mathbf{v}_j)$ for answer $a$ and the $i$-th and $j$-th object features $\mathbf{v}_i$ and $\mathbf{v}_j$ as the amount of sensitivity that $\mathbf{v}_j$ surpasses $\mathbf{v}_i$, as shown in Eq. 2.

$$\mathcal{SV}(a, \mathbf{v}_i, \mathbf{v}_j) = \max\left(\mathcal{S}(a, \mathbf{v}_j) - \mathcal{S}(a, \mathbf{v}_i), 0\right) \tag{2}$$

Based on the assumption that the proposal set contains at least one influential object that a human would use to infer the answer, we impose the constraint that the most sensitive object in the proposal set should not be less sensitive than any object outside the proposal set. Therefore, we introduce the influence strengthen loss $\mathcal{L}_{infl}$ in Eq. 3:

$$\mathcal{L}_{infl} = \min_{\mathbf{v}_i \in \mathcal{I}} \left( \sum_{\mathbf{v}_j \in \mathcal{V} \setminus \mathcal{I}} \mathcal{SV}(a_{gt}, \mathbf{v}_i, \mathbf{v}_j) \right) \tag{3}$$

where the $a_{gt}$ denotes the ground truth answer. The key differences between our influence strengthen loss and the ranking-based HINT loss are that (1) we relax the unnecessary constraint that the objects should follow the exact human ranking, and (2) it is easier to adapt to different types of explanation (*e.g.* textual explanations) where such detailed rankings are not available.

## 4.2 Criticizing Incorrect Dominant Answers

Next, for the incorrect answers ranked higher than the correct answer, we attempt to decrease the sensitivity of the influential objects. For example, in VQA-CP, bedrooms are the most common room type. Therefore, during testing, systems frequently incorrectly classify bathrooms (which are rare in the training data) as bedrooms. Since humans identify a sink as an influential object when identifying bathrooms, we want to decrease the influence of sinks on concluding bedroom.

In order to address this issue, we design a self-critical objective to criticize the VQA systems' incorrect but competitive decisions based on the most influential object $\mathbf{v}^*$ to which the correct answer is most sensitive as defined in Eq. 4.

$$\mathbf{v}^* = \arg\min_{\mathbf{v}_i \in \mathcal{I}} \left( \sum_{\mathbf{v}_j \in \mathcal{V} \setminus \mathcal{I}} \mathcal{SV}(a_{gt}, \mathbf{v}_i, \mathbf{v}_j) \right) \tag{4}$$

Specifically, we extract a bucket of at most $B$ predictions with higher confidence than the correct answer $\mathcal{B} = \{a_1, a_2, ..., a_{|\mathcal{B}|}\}$ and utilize the proposed self-critical loss $\mathcal{L}_{crit}$ to directly minimize the weighted sensitivities of the answers in the bucket $\mathcal{B}$ to the selected most influential object, as shown in Eq. 5.

$$\mathcal{L}_{crit} = \sum_{a \in \mathcal{B}} w(a)(\mathcal{S}(a, \mathbf{v}^*) - \mathcal{S}(a_{gt}, \mathbf{v}^*)) \tag{5}$$

where $a_{gt}$ denotes the ground truth answer. Because several answer candidates could be similar (*e.g.* *cow* and *cattle*), we weight the sensitivity gaps in Eq. 5 by the cosine distance between the answers' $300$-$d$ Glove embeddings [19], *i.e.* $w(a) = cosine\_dist(Glove(a_{gt}), Glove(a))$. In the multi-word answer case, the Glove embeddings of these answers are computed as the sum of the individual word's Glove embeddings.

### 4.3 Implementation and Training Details

In this section, we describe the detailed implementation and training procedure of our self-critical approach to VQA using VQA-X explanations.

**Training Details.** We first pre-train our base UpDn VQA system on the VQA-CP training set using standard VQA loss $\mathcal{L}_{vqa}$ (binary cross-entropy loss with soft scores as supervision) with the Adam optimizer [16] for at most 20 epochs. As suggested in [27], the learning rate is fixed to 10e-3 with a batch size of 384 during the pre-training process, and we use $1,280$ hidden units in the base UpDn VQA system. Then, we fine-tune our system to recognize important objects using $\mathcal{L}_{vqa} + \lambda_{infl}\mathcal{L}_{infl}$ with a learning rate of 10e-5 for at most 15 epochs on the intersection of VQA-X and VQA-CP training set. We initialize the model with the best model from the pre-train stage. In this stage, we also find the best influence strengthening loss weight $\lambda_{infl}^{\star}$. Finally, we fine-tune the system with the joint loss $\mathcal{L} = \mathcal{L}_{vqa} + \lambda_{infl}^{\star}\mathcal{L}_{infl} + \lambda_{crit}\mathcal{L}_{crit}$ for at most 15 epochs with a learning rate of 10e-5 on the intersection of VQA-X and VQA-CP training set. The bucket size $|B|$ of the competitive answers is set to 5 because we observed that the top-5 overall score of the pre-trained system on the VQA-CP dataset achieves 80.4%, and increasing the bucket size only marginally improves the score.

**Implementation.** We implemented our approach on top of the original UpDn system. The base system utilizes a Faster R-CNN head [22] in conjunction with a ResNet-101 base network [10] as the object detection module. The detection head is pre-trained on the Visual Genome dataset [17] and is capable of detecting $1,600$ objects categories and $400$ attributes. UpDn takes the final detection outputs and performs non-maximum suppression (NMS) for each object category using an IoU threshold of $0.7$. Then, the convolutional features for the top 36 objects are extracted for each image as the visual features, *i.e.* a $2,048$ dimensional vector for each object. For question embedding, following [2], we perform standard text pre-processing and tokenization. In particular, questions are first converted to lower case and then trimmed to a maximum of 14 words, and the words that appear less than 5 times are replaced with an "<unk>" token. A single layer GRU [6] is used to sequentially process the word vectors and produce a sentential representation for the pre-processed question. We also use Glove vectors [19] to initialize the word embedding matrix when embedding the questions. The size of proposal object set is set to 6.

## 5 Experimental Results

First, we present experiments on a simple synthetic dataset to illustrate basic aspects of our approach. We then present experimental results on the VQA-CP (Visual Question Answering with Changing Priors) [1] dataset where the QA pairs in the training data and test data have significantly different distributions. We also present experimental results on the VQA v2 validation set for completeness. We compare our self-critical system's VQA performance with the start-of-the-art systems via the standard evaluation metric. After that, we perform ablation studies to verify the contribution of strengthening the influential objects and criticizing competitive answers. Finally, we show some qualitative examples to illustrate the effectiveness of criticizing the incorrect answers' sensitivity.

### 5.1 Results on Synthetic Data

We manually created a dataset where the inputs are drawn from a mixture of two Gaussians, *i.e.* $\mathcal{N}_1 = \mathcal{N}([-3,3]^T, 2I_2)$ and $\mathcal{N}_2 = \mathcal{N}([3,3]^T, 2I_2)$, where each distribution defines a category. In order to ensure the training and test data have different category distributions, we intentionally assign different weights to the two components. In particular, during training, the examples are drawn from $\mathcal{N}_1$ with probability $p$, and during test, the examples are drawn from $\mathcal{N}_1$ with probability $1 - p$. We examine the effectiveness of our self-critical approach varying $p$ from 0.05 to 0.5 (*i.e.* 0.05, 0.1, 0.2, 0.5) (0.5 means no train/test difference). In these experiments, we use the obvious human explanation that the first channel (x-axis) is important for all training examples. We use a 15-layer feed-forward neural network with 256 hidden units and 1000 examples for both training and test in all of our experiments. We use Adam to optimize our model with a learning rate of 1e-3 during pre-training (100 epochs) with binary cross-entropy loss, and 1e-5 during fine-tuning (50 epochs) with our self-critical approach. The influence strengthening loss weight and self-critical loss weight are set to 20 and 1000, respectively. The results in Fig. 3 shows that the self-critical approach helps

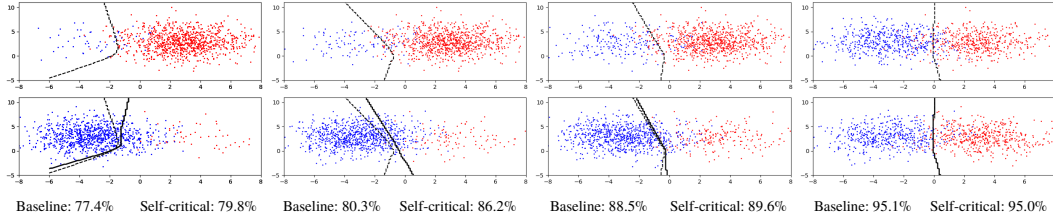

Baseline: 77.4%  Self-critical: 79.8%  Baseline: 80.3%  Self-critical: 86.2%  Baseline: 88.5%  Self-critical: 89.6%  Baseline: 95.1%  Self-critical: 95.0%

Figure 3: Decision boundaries and test set accuracies on synthetic data with various class ratios $p$, which is varied from 0.05, 0.1, 0.2, to 0.5 from left to right. The training data is shown in the top row, testing in the bottom. Red and blue colors denote different categories. Dashed lines and solid lines denote the boundaries of the pretrained and fine-tuned models, respectively.

| | Expl. | VQA-CP v2 test | | | | VQA v2 val | | | |
|---|---|---|---|---|---|---|---|---|---|
| | | All | Yes/No | Num | Other | All | Yes/No | Num | Other |
| GVQA[1] | | 31.3 | 58.0 | 13.7 | 22.1 | 48.2 | 72.0 | 31.2 | 34.7 |
| UpDn [2] | | 39.7 | 42.7 | 11.9 | 46.1 | 63.5 | 81.2 | 42.1 | 55.7 |
| UpDn+AttAlign [25] | | 38.5 | 42.5 | 11.4 | 43.8 | 61.0 | 78.9 | 38.4 | 53.3 |
| UpDn+AdvReg. [21] | | 41.2 | 65.5 | **15.5** | 35.5 | 62.8 | 79.8 | 42.4 | 55.2 |
| UpDn+SCR (ours) | QA | **48.47** | **70.41** | 10.42 | **47.29** | 62.3 | 77.4 | 40.9 | 56.5 |
| UpDn+HINT [25] | HAT | 47.7 | 70.0 | 10.7 | 46.3 | 62.5 | 80.5 | 41.8 | 54.0 |
| UpDn+SCR (ours) | HAT | 49.17 | 71.55 | 10.72 | 47.49 | 62.2 | 78.9 | 41.4 | 54.3 |
| UpDn+SCR (ours) | VQA-X | **49.45** | **72.36** | 10.93 | **48.02** | 62.2 | 78.8 | 41.6 | 54.5 |

Table 1: Comparison of the results on VQA-CP test and VQA v2 validation dataset with the state-of-the-art systems. The upper part includes VQA systems without human explanations during training, and the VQA systems in the bottom part use either visual or textual human explanations. The "Expl." column shows the source of explanations for training the VQA systems. SCR is the short hand for our self-critical reasoning approach. The results with a precision of 2 decimal points denote the mean of three runs with different random initial seeds.

shift the decision boundary towards the correct, unbiased position, increasing robustness and accuracy on the test data.

## 5.2 Results on VQA Data

**VQA Performance on VQA-CP and VQA v2 datasets**
Table 1 shows results on the VQA-CP generalization task, comparing our results with the state-of-the-art methods. We also report our system's performance on the balanced VQA v2 validation set for completeness.

Our system significantly outperforms other state-of-the-art system (e.g., HINT [25]) by $1.5\%$ on the overall score for VQA-CP when using the same human visual explanations (VQA-HAT), which indicates the effectiveness of directly criticizing the competitive answers' sensitivity to the most influential objects. Using human textual explanations as supervision is even a bit more effective. With only about half the number of explanations compared to VQA-HAT, these textual explanations improve VQA performance by an additional $0.3\%$ on the overall score, achieving a new state-of-the-art of $49.5\%$.

Without human explanations, our approach that only uses the QA proposal object set as supervision clearly outperforms all of the previous approaches, even those that use human explanations. We further analyzed the quality of the influential object proposal sets extracted from the QA pairs by comparing them to those from the corresponding human explanations. On average, the QA proposal sets contain 57.1% and 54.3% of the objects in the VQA-X and VQA-HAT proposal object sets, respectively, indicating a significant but not perfect overlap.

Note that our self-critical objective particularly improves VQA performance in the 'Yes/No' and 'Other' question categories; however, it does not do as well in the 'Num' category. This is understand-

| | Expl. | $\lambda_{infl}$ | $\lambda_{crit}$ | VQA-CP v2 test | | | |
|---|---|---|---|---|---|---|---|
| | | | | All | Yes/No | Num | Other |
| UpDn [2] | | | | 39.7 | 42.7 | 11.9 | 46.1 |
| UpDn+SCR (ours) | VQA-X | 5 | 0 | 46.6 | 62.2 | 12.1 | **47.9** |
| UpDn+SCR (ours) | VQA-X | 20 | 0 | **47.8** | **67.9** | **12.4** | 47.0 |
| UpDn+SCR (ours) | VQA-X | 60 | 0 | 47.2 | 65.0 | 11.9 | 47.6 |
| UpDn+SCR (ours) | VQA-X | 80 | 0 | 47.0 | 64.3 | 11.8 | 47.5 |

Table 2: Ablation study on various influence-strengthening loss weights on VQA-CP test data (. The "Expl." column shows the source of explanations for training the VQA systems. The "$\lambda_{infl}$" column shows the influence-strengthening loss weight. The "$\lambda_{crit}$" column shows the self-critical loss weight. SCR is the short hand for our self-critical reasoning approach.

| | Expl. | $\lambda_{infl}$ | $\lambda_{crit}$ | VQA-CP v2 test | | | |
|---|---|---|---|---|---|---|---|
| | | | | All | Yes/No | Num | Other |
| UpDn [2] | | | | 39.7 | 42.7 | **11.9** | 46.1 |
| UpDn+SCR (ours) | VQA-X | 20 | 500 | 48.7 | 70.0 | 10.4 | 47.8 |
| UpDn+SCR (ours) | VQA-X | 20 | 2000 | **49.5** | **72.2** | 11.0 | **48.1** |
| UpDn+SCR (ours) | VQA-X | 20 | 4000 | 49.4 | 71.9 | 11.0 | **48.1** |
| UpDn+SCR (ours) | VQA-X | 20 | 6000 | 49.0 | 71.6 | 10.9 | 47.8 |

Table 3: Ablation study on various self-critical loss weights on VQA-CP test data. The "Expl." column shows the source of explanations for training the VQA systems. The "$\lambda_{crit}$" column shows the self-critical loss weight. SCR is the short hand for our self-critical reasoning approach.

able because counting problems are generally harder than the other two types, and requires the VQA system to consider *all* of the objects jointly. Therefore, criticizing only the most sensitive ones does not improve the performance.

For the VQA v2 test dataset, our self-critical methods are competitive with previous approaches. This indicates that criticizing the wrong answers' sensitivities at least does not hurt performance when the training and test data have the same distribution.

**Ablation Study on the Loss Weights**
Tables 2 and 3 evaluate the impact of varying the weight of the influence strengthening loss and self-critical loss on the VQA-CP test data using VQA-X textual explanations. Table 2 shows that without $\mathcal{L}_{crit}$ to criticize the false sensitivity, our influence-strengthening still improves the UpDn VQA system $8.1\%$ on the overall score. As shown Table 3, combining with the $\mathcal{L}_{crit}$ loss, our approach sets a new state-of-the-art score ($49.5\%$) on the VQA-CP test set using textual explanations. We also notice that our approach is fairly robust to changes in the weight of both losses $\mathcal{L}_{infl}$, $\mathcal{L}_{crit}$ and consistently improves VQA performance for a wide range of loss weights.

**Study on Proposal Influential Object Set Size**
Table 4 reports results with various set sizes indicating the two objectives are fairly robust. We use VQA-HAT visual explanations to construct the influential object sets and both losses to fine-tune our model.

| $|\mathcal{I}|$ | 4 | 5 | 6 | 7 | 8 | 10 |
|---|---|---|---|---|---|---|
| VQA-CP v2 test | 48.8% | 49.1% | 49.2% | 49.1% | 48.7% | 48.3% |

Table 4: Ablation study on the size of the proposal influential object set.

**Effectiveness of Criticizing False Sensitivity**
In this section, we quantitatively evaluate the effectiveness of the proposed self-critical objective. In particular, we evaluate the fraction of false sensitivity where the predicted incorrect answer's sensitivity to the influential object (to which the correct answer is most sensitive) is greater than the

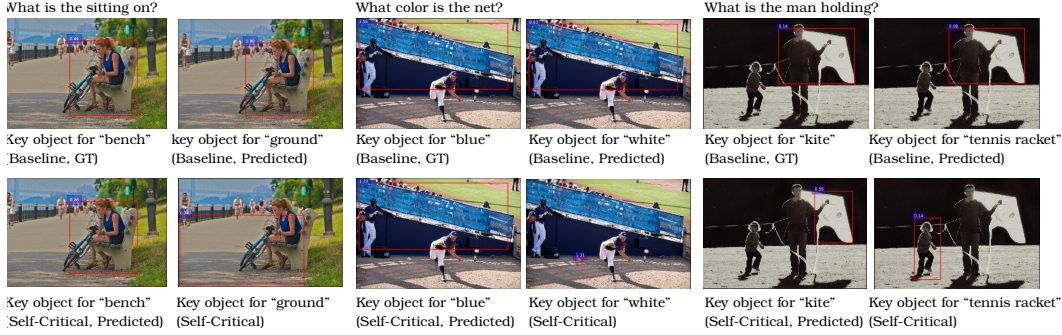

Figure 4: Positive examples are showing that our self-critical reasoning approach prevents the incorrectly predicted answer in the UpDn baseline system from being sensitive to the most influential object. For each example, the top two figures show the object to which the ground truth (left) and incorrectly predicted (right) answers are sensitive. The bottom two figures show the corresponding most influential object after our self-critical training. Note that the attention for the incorrect answer shifts to a more relevant part of the image for that answer. The number around the bounding box is the answer's sensitivity to the object.

correct answer's sensitivity. We formally define the false sensitivity rate in Eq. 6:

$$\mathcal{FSR} = \frac{\sum_{Q,\mathcal{V}} \mathbb{1}[\mathcal{S}(a_{pred}, \mathbf{v}^*) - \mathcal{S}(a_{gt}, \mathbf{v}^*) > 0, score(a_{pred}) = 0]}{\sum_{Q,\mathcal{V}} 1} \quad (6)$$

where $\mathbb{1}[\cdot]$ denote the function that returns 1 if the condition is satisfied and returns 0 otherwise.

For the original UpDn VQA system, we observe a false sensitivity rate of 35.5% among all the test QA pairs in the VQA-CP. After the self-critical training, the false sensitivity rate reduces to 20.4% using the VQA-HAT explanations, and to 19.6% using VQA-X explanations. This indicates that false sensitivity is a common problem in VQA systems and shows the utility of addressing it.

Some examples of how our self-critical approach mitigates false sensitivity are shown in Figure 4. Note that for the correct answer, our approach increases the influence of the most influential object, which we attribute to the influence strengthening part. More importantly, we observe that this object's influence on the *incorrect* answer *decreases* and sometimes falls below other objects.

| | UpDn | UpDn + QA | UpDn + HAT | UpDn + VQA-X |
|---|---|---|---|---|
| $\mathcal{FSR}$ | 35.5% | 22.6% | 20.4% | 19.6% |

Table 5: False sensitivity rate ($\mathcal{FSR}$) comparison of using different types of human explanations.

## 6  Conclusion and Future Work

In this work, we have explored how to improve VQA performance by criticizing the sensitivity of incorrect answers to the most influential object for the correct answer. Our "self-critical" approach helps VQA systems generalize to test data where the distribution of question-answer pairs is significantly different from the training data. The influential objects are selected from a proposal set extracted from human visual or textual explanations, or simply from the mentioned objects in the questions and answers. Our approach outperforms the state-of-the-art VQA systems on the VQA-CP dataset by a clear margin even without human explanations as additional supervision. In the future, we would like to combine the visual and the textual explanations together to better train VQA systems. This is difficult because the proposal object sets for these two types of explanations contain different types of noise (i.e., question-irrelevant objects), and therefore different biases.

## Acknowledgement

This research was supported by the DARPA XAI program under a grant from AFRL.

## Footnotes

[1]We exam these situations by designing a metric called false sensitivity rate ($\mathcal{FSR}$) in Sec. 5.2.

[2]The key approach used by the VQA-challenge winning entries in the last two years.

[3] $\mathbf{1}$ denotes a vector with all 1's.

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
