[Reviews · NeurIPS 2019]

Reviewer 1



Originality: The proposed work is inspired from an existing work – HINT (Selvaraju et al., arXiv 2019) which also proposes a novel training objective to align gradient based model’s importance for various object proposals in the image with the regions identified as important by humans. This paper improves upon HINT by – 1) instead of training the model to align its gradient based importance with regions identified as important by humans, the paper trains the model to strengthen its importance for the most influential region -- proposal deemed as most important as per the model’s gradients based importance among the set of regions identified as most important by humans, 2) in addition to using visual regions identified as important by humans, the paper also introduces using textual explanations provided by humans and training QA pairs to identify important image regions, 2) the paper proposes another term in the objective that criticizes incorrect predicted answers being more sensitive to the influential region than correct answers. Quality: The paper does a good job of evaluating the proposed approach on both the VQA-CP and VQA datasets. The evaluation of the ablations of the proposed approach and false sensitivity rate are also useful. Clarity: The paper is clear for the most part except the following – 1. Currently, in order to understand how the gradients from the proposed training objectives are effecting the model’s parameters, one needs to read the HINT paper. It would be great if this paper also describes how the sensitivity is a function of the models’ parameters, for the convenience of the readers. 2. The premise of the self-critical loss is that the correct answer should be more sensitive than the incorrect answers to the influential region. However, how does this apply to all those questions which are about identifying attributes in a specific image region, such as, “What color is the person’s shirt?”. For such questions, even for incorrect colors, the model still needs to look at the same shirt region that a model should look at for predicting correct colors. Significance: The improvements obtained using the proposed approach on the VQA-CP dataset are significant. The proposed novel training objectives and the ideas of using human textual explanations and training QA pairs to obtain important region regions have the potential to open-up new directions of thinking about enforcing visual grounding. ---- Post-rebuttal comments ---- The authors have responded to my concerns satisfactorily. I am happy to increase the score to 8.

Reviewer 2



The authors tackle generalization issues in VQA tasks that may raise due to unbalanced datasets. They propose two fine-tuning losses to better link key objects to ground truth answers to reduce answers biases, namely: - The influence strengthening loss. It ensures that it exists at least one potential influential object (which are predetermined with well-motivated heuristics) that lead to the correct answer - The self-critical loss. It ensures that the most influential object is correctly tied to the right answer by reducing its sensitivity to other false-negative answers. In other words, the full method assumes that visual question can be answered by relying on one prominent object. Even if I think it is a strong assumption, it is in line with previous works on VQA-CP. However, It would have been very beneficial to provide additional arguments to support this choice. For instance, the authors mentioned that they observe a recurrent discrepancy between valid object attention masks and incorrect answers. However, they do not provide global metrics to assess their claim and they only give one example in Figure 1. We need to look at figure 4 at the end of the paper to finally ground the claim to facts. Differently, it would have been highly appreciated to highlight the interests and weaknesses of this approach (e.g., it does not take into account language compositionally). I appreciate that the authors used different methods to extract influential objects: Human attention (in line with previous works), text explanation (to rely on another modality), and question parsing (to remove the need of extra annotation). As a complementary analysis, I would have compared object sets (Jaccard Distance) which are extracted with visual cues and text description. Indeed, the VQA-X dataset contains both information for each question/answer pairs. The method is more or less correctly explained. The training details seems complete and allow for reproducibility. The authors do not provide code source although they mentioned it in the reproducibility checklist. The empirical results are quite convincing and the necessary baselines and ablation studies are correctly provided. The formatting is simple and clear! It would have been perfect to provide the error bar as the number of experiments remains low (and over a small number of epochs) The cherry on the cake would be to run similar experiments on VQAv1 / VQA-CP1? To increase the impact of the paper, I would recommend extending the setting to either dense image captioning, or question answering (if possible). I feel that the discussion section raise some excellent points: - I really like table 4, that clearly show that the method perform as expected (I would have add HINT for being exhaustive) - the ablation study is convincing But, a lot of open-questions are still left open and could have been discussed. For instance, I would have appreciated a more in-depth analysis of model errors. What about the model complexity? Why only reweighting L_{crit}. How does evolve L_crit and L_infl at training time? On a more general note, I think the overall writing and paper architecture can be greatly improved. For instance, - the introduction and related work can be partially merged and summarized. - 4.2 starts by providing high-level intuition while 4.1 does not. - Training details incorporate some result discussion Generic questions (sorted by impact): - What is the impact of |I|, do you have the performance ration according to the number of top |I| influential objects - Eq1 is a modified version of GardCAM, however, the modifications are not highlighted (neither explained). For instance, why did the authors remove the ReLU - Even if the weight sensitivity in equation 5 is well motivated, it is not supported by previous works. Thus, did you perform an ablation study? It would be very have been nice in the discussion section. - What is the actual computation cost of the two losses? What is the relative additional time required? +5%, +20%, +200%? - As you used heuristics to retrieve influential objects, did you try to estimate the impact of false negatives in the loss. - How did you pick 0.6 for glove embedding similarity? Did you perform k-cross-validation? What is the potential impact - Have you tried other influential loss (Eq3)? For instance, replacing the min with a mean or NDCG? Remarks: - I would use a different notation for SV(.,.,.) as it is not symmetric. For instance SV_{a}(v_i || v_j) would avoid confusion (I am using KL notation here) - Non-formal expression should be avoided: Ex: "l32 What's worse" - The references section is full of format inconsistencies. Besides, some papers are published with proceeding but are referred to arxiv papers. - 3.1 introduces non-important notation, e.g., function h(.) or f(.) that are never used in the paper. - Several subsections could be gathered together, or define as a paragraph: 2.1/2.2/2.3 ; 5.1/5.2/5.3, etc. It would have save space for more experiments Conclusion: The paper introduces two losses to better tie influential objects and potential answers. The method is convincing, and the experimental results are diverse and good. However, I still think that the paper requires further polishing to improve the readability. I would also advocate providing more element to support the proposed method and to analyze the strengths and weaknesses. Although the current experiences are quite convincing, I would advocate adding more analysis to definitely conclude the efficiency of the method. ---------------------------------- The rebuttal was clearly written and insightful. it answered most of my questions, and the authors demonstrate their ability to update the paper accordingly. Therefore, I am happy to increase my score, and accept the paper

Reviewer 3



The paper proposes a method to perform VQA tasks using self-critical reasoning. The paper argues that the model should not only focus on the right/important reason when giving the correct answer, it should also not focus on the important regions while providing wrong answers. The main contribution of this work is to how not to focus on those important regions while providing wrong answers. Identification of important regions is a hard problem, so they reduced this to identifying influential objects in the image for the QA pairs. The influential objects are obtained using 3 methods: a visual explanation using VQA-HAT, textual explanations using Park et al and automatically by parsing QA pairs. Using selected influential objects, VQA model based on UpDn approach is proposed such that it penalizes the high ranked incorrect answers to make them less sensitive to influential objects. To do this, they propose an influence strengthen loss such that the correct answer is more sensitive to the corresponding object. To decrease the influence of the influential object on the incorrect answer, a self-critical loss is proposed. Author's have provided the utility of the method on the generalizable VQA-CPv2 and VQAv2 dataset. Results clearly show that the proposed method outperforms other methods, on the VQA-CP dataset and comparable performance on the VQAv2 dataset. However, having a 0.3% improvement using the textual description is not that significant. The ablation study clearly shows that having a self-critical loss only improves performance by 6.3%, which is very significant. At the same time, having only influence strengthen loss improves the performance comparatively more. It will good to comment on that also. Specifically, Yes/No and Other type questions are more effected by influence loss. Do authors thought about having a significance test on these results and can show that both these loss performances totally different or similar manner? Overall the proposed method is novel. Some of the things which are not clear: - L25-30: says that "which we have found occurs quite frequently". How this "frequently" is being calculated. - L111-112: when authors say "we assume that it at least includes the most relevant object.", again do they did some sort of quantification of this, at least on the small subset - L127-131 authors describe automatic creation of proposal set. However, in experiments, it is not clear that all 3 is being used or only some of those. It will be good to have all 3 separately and then combinations. - For the VQAv2 dataset, how much results are comparable because it has extra data in terms of explanation data like HAT & VQA-X - Did the authors try to have a different weighting scheme for the loss? Minor -L111: maybe -The figure needs to be improved, slightly bolder box and the number is very hard to read -If available, please cite the published version of the paper e.g [27, 30]

[Author Response · NeurIPS 2019]

We gratefully thank all reviewers for their valuable comments. We will try our best to address them in the revision.

**Comments on Clarity for R #1.**
(1) Thanks for pointing out the missing details in HINT paper, we will try to add a description of how the gradient-based
sensitivity score effects the model parameters through second order gradients when defining our proposed objectives.
(2) For the identifying attributes case, even though the influential region should support both correct and incorrect
answers, it should *contribute more* to the correct one. The self-critical loss helps update model parameters to increase
the right answer's sensitivity and decrease the wrong ones to the influential object until the model is right.

**Comments on Measuring the Frequency of Wrong Answers with Valid Attention for R #2 & R #4.**
We agree that a global measurement makes our claims stronger. In particular, we think our false sensitivity rate defined
in Eq. 6, which computes the fraction of false sensitivity where the predicted incorrect answer's sensitivity to the
influential object is greater than the correct answer's sensitivity, can partially help indicate that the false sensitivity is a
common problem in VQA.

**Comments on the discrepancy of the influential objects extracted from different explanations for R #2.**
Thanks for the proposal of using Jaccard Distance to compare the object sets extracted with visual cues and text
description in VQA-X validation and test set. We will include this measurement in our revision.

**Comments on Significant tests for R #2 & R #4.**
Thanks for mentioning the significance-test issue, we believe it would make our results more convincing. We will add
error bars on the VQA scores to show the stabilities of our proposed objectives.

**Comments on Various Ablation Study Issues for R #2 & R #4.**
(1) *Influence Strengthening Loss and Self-Critical Loss*: Currently, we only show the results of using and not using the
influence strengthening loss due to the page limits and focus more on the self critical loss, since the former is more
intuitive and mentioned in HINT paper (Selvaraju et al., arXiv 2019). We hope to add the detailed ablation study on the
influence strengthening loss to the paper in the final version for both completeness and further analysis. Also, as pointed
out by Reviewer # 4, in order to analyze the behavior of both losses, we will measure the intersection over union (IOU)
between the sets of QA pairs that are correctly answered trained with one of the two objectives but incorrectly answered
using simply the original VQA objective.
(2) *Proposal Set Size $|\mathcal{I}|$*. Thanks for pointing out the missing details on $|\mathcal{I}|$. We set it to 6 in all of our experiments in
the submission. Table. 1 reports the ablation results with various set sizes indicating the two objectives are fairly robust.
We use VQA-HAT visual explanations to construct the influential object sets and both losses to fine-tune our model.

| $|\mathcal{I}|$ | 4 | 5 | 6 | 7 | 8 | 10 |
|---|---|---|---|---|---|---|
| VQA-CP v2 test | 48.8% | 49.1% | 49.2% | 49.1% | 48.7% | 48.3% |

Table 1: Ablation study on the size of proposal influential object set.

**Comments on Generic Questions for R #2.**
*(1) Modified Version of GradCAM.* We use the same methods to compute the sensitivity score adopted in the HINT paper
and we also empirically find that this modified GradCAM, which removes the ReLU, works better than original one.
Our interpretation is that before ReLU, the negative values could provide counter-factual meanings to the prediction.
Therefore, there is no need to filter them out using ReLU. However, in the original GradCAM visualization paper was
more focuesd on visualizing the regions that positively contribute to the predictions.
*(2) Model Complexity and Computation Cost.* First, our model does not introduce parameters to the original VQA
systems, and therefore the model parameter complexity remains the same. During forwarding and backwarding, original
VQA system takes about 20min per epoch and with the two objectives the system takes about 25min using a TITAN V.
*(3) False Negatives.* Thanks for mentioning the false negative issue in the model. We think the impact of this issue is
hard to evaluate since there are no gold standard object annotations. However, it is partially revealed by the ablation
study on the proposal set size $|\mathcal{I}|$ in Table. 1, *e.g.* larger size reduces false negatives but increases false positives.
*(4) Glove Embedding Similarity.* We did not do k-cross-validation as we do not have gold standard annotations on
which word should be associated with which object and we simply inspect some examples manually.
*(5) Other Influential Loss.* We tried "mean" as the influential loss and found "min" works better. Our interpretation is
"min" only requires at least *one* object to be influential while "mean" encourages all of the objects in the proposal object
set to be influential, which is more vulnerable to false positives in the proposal set.

**Comments for R #4.**
*(1) Quantification of L111-112.* For visual explanations, we can ensure that the proposal set contains the regions that
humans find most important. When using textual explanations and QA pairs, we will use Jaccard Distance (mentioned
by R #2 ) to measure the similarity between the proposal sets in VQA-X validation and test set in our final version.
*(2) Usage of Proposal Set.* We currently use the three kinds of proposal sets separately denoted under "Expl." in tables.

[Meta-Review · NeurIPS 2019]

All reviewers recommended the submission for acceptance. Reviewers found the author response to be insightful and helped clarify many of their concerns. The approach itself introduces an interesting take on bias reduction for VQA that proves effective across a range of experimental settings.